# Placental Characteristics Classification of Various Native Turkish Sheep Breeds

**DOI:** 10.3390/ani11040930

**Published:** 2021-03-25

**Authors:** Uğur Şen, Hasan Önder, Emre Şirin, Selçuk Özyürek, Dariusz Piwczynski, Magdalena Kolenda, Sezen Ocak Yetişgin

**Affiliations:** 1Department of Agricultural Biotechnology, Ondokuz Mayis University, Samsun 55139, Turkey; 2Department of Animal Science, Ondokuz Mayis University, Samsun 55139, Turkey; honder@omu.edu.tr (H.Ö.); sezen.ocak@omu.edu.tr (S.O.Y.); 3Department of Agricultural Biotechnology, Kırşehir Ahi Evran University, Kırşehir 40100, Turkey; emre.sirin@ahievran.edu.tr; 4Department of Veterinary, Erzincan Binali Yıldırım University, Erzincan 24500, Turkey; sozyurek@erzincan.edu.tr; 5Department of Animal Biotechnology and Genetics, UTP University of Science and Technology, 85-084 Bydgoszcz, Poland; darekp@utp.edu.pl (D.P.); kolenda@utp.edu.pl (M.K.)

**Keywords:** sheep breeds, placental characteristics, hierarchical clustering, nearest neighbours method, principal component analyses

## Abstract

**Simple Summary:**

The aim of this study was to classify placental characteristics of Akkaraman, Morkaraman, Karayaka, Awassi, Malya, and Bafra native sheep breeds using the hierarchical clustering method. As a result, six breeds were separated into three clusters: the first cluster consisted of Bafra, Karayaka, and Awassi breeds; the second consisted of Akkaraman and Malya breeds; and the third cluster included only the Morkaraman breed.

**Abstract:**

The aim of this study was to classify placental characteristics of Akkaraman, Morkaraman, Karayaka, Awassi, Malya, and Bafra sheep breeds using the hierarchical clustering method. In total, 240 individual data records were used as experimental material. Placental characteristics such as total cotyledon surface area, small and large cotyledon length, small cotyledon depth, etc. were used as explanatory variables to classify the breeds’ characteristics. Hierarchical clustering was used with the nearest neighbour method with Euclidean distance in order to classify the sheep breeds’ variations. As a result, six breeds were separated into three clusters: the first cluster consisted of Bafra, Karayaka, and Awassi breeds; the second consisted of Akkaraman and Malya breeds; and the third cluster included only the Morkaraman breed. Bafra and Karayaka were pointed as the nearest breeds, with a similarity of 98.7% in terms of placental characteristics. The similarity rate of the Akkaraman and Malya breeds was at a level of 97.5%, whereas it was 96.8% for Bafra, Karayaka, and Awassi breeds. The similarity of Akkaraman, Karayaka, Awassi, Malya, and Bafra sheep breeds was estimated as 95.7%. The overall similarity was found to be at a level of 93.2% among sheep breeds. The outcomes of the study might be useful as a selection tool for reproductivity and can be used to select the breed to be reared.

## 1. Introduction

Small ruminants have been important components of rural life and still play a substantial role in the livelihood of farmers [1]. In Turkey, sheep production plays an important role in the livestock sector because of the country’s geography and climate, as well as social, cultural, and economic structures [2]. As in many sheep-producing countries, sheep are generally reared for lamb meat production. Reproductive characteristics have great importance for lamb production, as well as meat production and its quality. Various sheep breeds are used for meat production in developing countries, and most of them are local breeds that have adapted to the region where they are raised. There are more than 42 million sheep in Turkey, which are mainly fat-tailed breeds, and also a small percentage of crossbreeds with foreign genotypes [3]. Akkaraman, Morkaraman, Karayaka, Awassi, Malya, and Bafra sheep, covering almost 80% of native sheep breeds in Turkey [3], are largely suited to a variety of harsh geographic and environmental conditions [2,4,5,6]. Lambs of these breeds are an important source for the red meat sector under harsh climate conditions [4,5,6]. The reproductive performance and maternal ability of these native breeds have been improved recently, yet there are still some gaps in the knowledge of how placental efficiency affects foetal development. Therefore, it is essential to identify placental efficiency as a key tool to increase the reproductive efficiency and productivity of the native sheep breeds. Information about the similarity of breeds based on characteristics of interest can be used as a decision support tool in selecting the breed to be produced.

The structure of the placenta in sheep is multiple cotyledonary, this structure allows the placenta, where the foetus is located, to attach to the uterus endometrium [7]. Therefore, placental growth, development, and its nutrient transfer capacity play a central role in determination of the prenatal growth trajectory of the foetus, resulting in alteration of birth-related traits such as birth weight, birth type, and litter size [7,8]. Placental characteristics are a valuable indicator of offspring mortality for small ruminants [7,8]. Previous studies reported that lamb survival is highly correlated with birth weight [9], especially in multiple births, which are associated with a high risk of preterm birth and low birth weight [10]. Additionally, lambs born with low weight are slow to stand and suck, as their low weight is associated with poor postnatal viability [11,12]. Dwyer et al. [13] suggested that there may be variations between sheep breeds in terms of dividing nutrients into developing foetuses, and that there may be a difference in adaptation to deal with the poor feeding conditions encountered in sheep breeds. Identifying similarities in placental characteristics among breeds may indicate which breed has a higher reproductive performance and can adapt better to deal with the poor feeding conditions. Thus, placental properties can be used as selection criteria in endeavours undertaken to improve reproductive performance.

Hierarchical cluster analysis is a commonly used method of exploratory data analysis, the purpose of which is to group and classify categories based on their similarity to the explanatory variables [14]. In order to increase the reliability of the results of a cluster analysis, only effective variables should be included in the dataset. Principal component analysis is a method used to determine the effective variables according to their rate of explanation on the total variance [15]. Principal component analysis is generally used as a variable selection method for other multivariable methods such as clustering and discriminating. Therefore, cluster and principal component analyses could be used to provide a tool to build relationships that allow the clustering of animals by similar reproductive characteristics and correlations among different traits evaluated [16]. Previous studies reported that multivariate analyses such as cluster and principal component analyses are useful for addressing relevant decisions in animal breeding programs [17,18]. Multivariate analyses such as principal component analysis and hierarchical clustering analysis were used with great success on livestock data to show similarity or dissimilarity; in some, linear body size was used to differentiate among breeds and to calculate the genetic Mahalanobis distance for goats in Burkina Faso [19], body size was also used to differentiate local sheep, goat and duck breeds in Indonesia [20]. Another technique that has been used to estimate the genetic distance and diversity is blood protein polymorphisms on indigenous fat-tailed sheep populations [21]. DNA markers were used to describe the goat genetic distance and diversity [22,23], and the sheep breeds’ [24,25] mitochondrial DNA sequence was reported to calculate the genetic distance in sheep [26], sheep breeds were also classified according to wool profiles [27]. Hierarchical clustering was also successfully used to classify 16 terminal breeds such as Hampshire, Suffolk, Dorset and Rambouillet by SNP data [28].

Although many studies have been conducted on the relationship of placental characteristics to birth weight and birth-related traits in different sheep breeds, to our knowledge, there are no reports about the classification of sheep breeds in terms of placental characteristics and birth-related traits. This research was therefore carried out for the purpose of classifying Akkaraman, Morkaraman, Karayaka, Awassi, Malya, and Bafra native sheep breeds using a hierarchical clustering method based on their placental characteristics and birth-related traits.

## 2. Materials and Methods

### 2.1. Material

The experimental procedures were approved by the Local Animal Care and Ethics Committee of Kirsehir Ahi Evran University, Kirsehir, Turkey, ensuring compliance with EC Directive 86/609/EEC for animal experiments (approval number: 68429034/14). The study included the following native sheep breeds, which were in at least second parturition and ranged from 2–3 years of age: Akkaraman (*n* = 45), Morkaraman (*n* = 45), Malya (*n* = 40), Awassi (*n* = 40), Karayaka (*n* = 30), and Bafra (*n* = 40). The study was conducted within the normal seasonal breeding cycle of ewes (September to March), and each breed was raised under different geographical locations of Turkey. The Akkaraman and Malya breeds were obtained from the Malya farm of the General Directorate of Agricultural Enterprises (TİGEM) in Kirsehir (39°18′ N 34°20′ E), the Awassi from Ilci–Cicekdagi Agriculture Enterprise farm in Kirsehir (39°39′ N 34°25′ E), the Morkaraman from a private farm in Erzincan (39°80′ N 40°43′ E), the Karayaka from a private farm in Tokat (40°42′ N 36°35′ E), and the Bafra were obtained from the experimental sheep farm of Ondokuz Mayis University in Samsun (41°21′ N 36°10′ E). All breeds were raised under semi-extensive systems, and all farms where placentas were provided had similar management and feeding conditions. Also, all breeds were housed and cared for under similar conditions in the barns (research farms belonging to the state maintained identical conditions, and commercial farms were selected by their similarity of management practices). The ewes were allowed to graze 5 h/day during gestation. Additionally, the ewes were fed with the concentrate (at least 90% dry matter, 18% crude protein, and 2700 kcal/kg metabolic energy) on the average at the level 100 g/ewe/day and wheat straw (at least 91% dry matter, 3% crude protein and 1400 kcal/kg metabolic energy) at 1 kg/ewe/day during the last third of gestation.

### 2.2. Methods

After the lambing, ewes and their lambs were left undisturbed for a period to allow sufficient time for a bond to be established between the mother and offspring. The birth-related traits (birth type, lamb’s birth weight, and gender) were recorded within 12 h after parturition. The lambs were weighed using suspended scales (range 0–20 kg) and recorded. Each ewe was left to deliver the placenta naturally, and placentas were collected from singleton gestations immediately after delivery; care was taken to ensure that any placental weights (PWs) taken were of the total placenta with all fluid removed before weighing. The total cotyledon numbers (TCNs) and total cotyledon weights (TCWs) of placental cotyledons dissected from the chorioallantois were also counted and determined. Average cotyledon length (ACL), depth (ACDe), width (CWi), and volume (CV) were measured with an electronic digital compass selecting thirty cotyledons for each size groups (small, <20 mm diameter; medium, 20–30 mm diameter; large, >30 mm diameter). The total cotyledon surface area was calculated after taking the measurements of all cotyledons in individual placentas as (cm^2^) with the following formula: radius squared of cotyledon [((CWi + CL)/4)^2^] × 3.14 (π) × TCN. Additionally, placental efficiency (PE; lamb BW/placental weight), cotyledon efficiency (CE; lamb BW in grams/total cotyledon surface area), volumetric cotyledon efficiency (VCE; cotyledon volume/placental weight), and cotyledon density (CD; number of cotyledons/per gram placental weight) were calculated for each ewe [8,29].

The explanatory variables used to classify the breeds included total cotyledon surface area, small cotyledon length, medium cotyledon length, large cotyledon length, small cotyledon depth, medium cotyledon depth, large cotyledon depth, small cotyledon width, medium cotyledon width, large cotyledon width, cotyledon volume, total cotyledon number, small cotyledon number, medium cotyledon number, large cotyledon number, cotyledon efficiency on volume, placental weight, placental efficiency, cotyledon efficiency, cotyledon density, litter weight, birth type, and gender. Principal component analysis was used to determine the effective variables on total variance. Principal component analysis can be applied to any group of variables to summarize the highest possible variability in a reduced number of variables (principal components) [30]. Principal component analysis can be used with great robustness for many types of data for dimension reduction [31]. To classify the sheep breeds, a hierarchical clustering algorithm was used with the nearest neighbourhood method with Euclidean distance. Hierarchical clustering analysis according to the factor scores was derived from principal component analysis [30]. Tree-based hierarchical clustering of individuals to define clusters of similar populations according to interested traits was strongly suggested [32]. A dendrogram is structured where the root corresponds to a cluster containing all data points and the leaves correspond to the *n* input data points. Each internal node of the dendrogram corresponds to a cluster of the data points in its sub-tree. The clusters (internal nodes) become more refined as the nodes are lower in the tree. The goal is to construct the tree so that the clusters deeper in the tree contain points that are relatively more similar [33]. Analysis of variance (ANOVA) was used to analyse the reproductive traits of sheep breeds, and Duncan’s multiple comparison tests were used to examine the differences of the means. All analysis was executed using R software with *FactoMineR* and *factoextra* packages [34].

## 3. Results

The main objective of this study was to conduct the classification of sheep breeds using a hierarchical clustering, exploring relationships and dependencies of placental characteristics that significantly affect the reproductive function of sheep. This investigation was based on data collected from Akkaraman, Morkaraman, Karayaka, Awassi, Malya, and Bafra sheep breeds raised in several farms around regions of Turkey, using principal component and hierarchical clustering analyses. Results of these statistical methods and concepts of this study obtained by using principal component and hierarchical clustering analyses may be applicable to the Sheep and Goat Breeders’ Association of Turkey, which have data collected on large volumes of the placental traits of native sheep breeds. The sample size of this study was enough for placental and birth-related traits, according to the power of the test which was obtained as 88.73%. This finding showed that the results can be generalized to the population with high robustness.

The reproductive traits of Akkaraman, Morkaraman, Karayaka, Awassi, Malya, and Bafra sheep breeds are given in Table 1. There were no significant differences among breeds in terms of fertility rate. However, litter size at birth of the Malya, Awassi, and Akkaraman breeds were found to be higher than the other breeds.

Descriptive statistics of placental characteristics and birth-related traits in the Akkaraman, Morkaraman, Karayaka, Awassi, Malya, and Bafra sheep breeds are given in Table 2. According to the results, the effect of breeds on placental characteristics were found to be statistically significant (*p* < 0.05).

Contribution of components on explained total variance was present in Figure 1. Contribution of components for first, second, third, fourth, fifth, sixth, and seventh were found as 30.6%, 17.7%, 12.3%, 10.8%, 7.3%, 4.6%, and 4%, respectively. Contribution of variables to components 1 and 2 were present in Figure 2. The most effective explanatory variable was the total cotyledon surface area (8.1%), and other effective variables were small cotyledon length (7.1%), cotyledon volume (7.03%), small cotyledon width (6.84%), small cotyledon depth (6.27%), total cotyledon number (6.12%), cotyledon efficiency on volume (5.33%), cotyledon density (4.87%), and placental weight (4.48%) according to contribution of variables to components 1 and 2. Contribution of all variables was found less than 10%.

Variance of breeds for contributing to components 1 and 2 are present in Figure 3. When the contribution of the placental characteristics of breeds on components 1 and 2 was examined, the variance of the Morkaraman breed in terms of the placental characteristics differed from other breeds with the position on the graph. The variance of the Malya breed in terms of the placental characteristics showed the greatest value. The placental characteristics of the Karayaka and Akkaraman breeds showed the smallest variance, that was more reliable than other breeds. These findings were proof of the reliability of clustering results with the use of descriptive statistics given in Table 2 that the standard errors were at acceptable levels.

The dendrogram showing the clustering of the Akkaraman, Morkaraman, Karayaka, Awassi, Malya, and Bafra sheep breeds is given in Figure 4. Clustering results indicated that the six breeds were separated into three clusters. The first cluster consisted of the Bafra, Karayaka, and Awassi breeds, the second cluster consisted of the Akkaraman and Malya breeds, and the third cluster included only the Morkaraman breed. The nearest breeds were estimated as the Bafra and Karayaka with a similarity of 98.7%. Moreover, the similarity of the Bafra, Karayaka, and Awassi breeds was estimated as 96.8% due to the breeds included in the first cluster. The similarity rate of the Akkaraman, Karayaka, Awassi, Malya, and Bafra was estimated as 95.7%. The similarity rate of Akkaraman and Malya breeds was indicated as 97.5%. The overall similarity was found to be at the level of 93.2% among the sheep breeds.

## 4. Discussion

Small cotyledon variables had a valuable contribution for explaining total variance. It may be caused from the statistical fact that these traits had a small effect on calculated traits such as total cotyledon surface area, cotyledon volume, and cotyledon density—that is, multicollinearity was not significant. The line plots in Figure 1 show the percentage of explained variance for the first ten principal components for the placental characteristics. According to principal component analysis results, the first (30.6%) and second (17.7%) components (dimensions) explain 48.30% of total variance. Necessary explanation rate [35] can be reached by using seven components by the value of 87.35%.

Dwyer and Lawrence [36] reported that different sheep breeds (hill and mountain) carry various weights of offspring during gestation as a proportion of their own body weight. These differences suggest that different breeds of ewes may represent different efficiency at partitioning nutrients during gestation for foetus development due to placental functional ability. The cotyledon surface area is an important indicator of the functional ability of the placenta, and previous studies suggested that determining cotyledon efficiency by measuring the cotyledon surface area is a far more conclusive and reliable indicator of placental functional ability [8,29]. This opinion is also supported by our findings that the contribution of variables depending on total cotyledon surface area was found more explanatory in the present study (Figure 2). Therefore, the cotyledon surface area is suggested to be used as an important marker in breed classification for foetal development during gestation.

A previous study investigated that the Akkaraman and Morkaraman breeds were found close to each other, while the Karayaka breed was found to be distant from them, according to 31 microsatellite markers [37]. The Akkaraman, Morkaraman, and Awassi breeds were found to be significantly close to each other for mitochondrial D-loop region [38]. The Awassi breed was found distant from the Akkaraman and Morkaraman breeds, according to 12 RAPD primers [39], which were different according to result of placental characteristics. Overall similarity was found to be 98.8% on the Mahalanobis distance of the Barbados Black Belly Cross, Garut Local, Garut Composite, Sumatra Composite, and St. Croix Cross sheep breeds based on body measurements [20]. Classifying of 16 terminal sheep breeds of USA was separated into six clusters according to SNP, and overall similarity was found to be 99% [28]. The close clustering of East African sheep populations and distinct separation from their northern counterparts was well demonstrated by phylogenetic, principal component, and cluster analyses. A total of 18 sheep breeds were separated into 10 clusters with an overall similarity of 94.62% [31]. The clustering method was used for typical microbial population features corresponding to different sheep breeds [40]. When the literature was interpreted, it was clear that when the overall similarity among sheep breeds was high, even though the examined traits are different. In that case, the chosen statistical method is important to reach the robust estimations. Hierarchical clustering analysis used with the nearest neighbour method with Euclidean distance executed only selected effective variables in the present study. As expected, the Bafra and Karayaka breeds were grouped in the same cluster due to their higher similarity ratio (98.7%), Bbecause the Bafra breed is a crossbreed of Karayaka with Chios sheep breeds [2,6]. Similarly, the Akkaraman and Malya breeds had a higher similarity ratio (97.5%) and were grouped in the same cluster. The Malya and Akkaraman sheep breeds being in the same cluster was also expected, since the Malya breed is a crossbreed of the Merino and Akkaraman breeds [5,41]. It is interesting that the Morkaraman breed was found most extreme in relation to other breeds, and the Awassi breed was placed in the cluster with the Bafra and Karayaka breeds even though Awassi is a fat-tailed sheep.

## 5. Conclusions

Placental characteristics of six examined sheep breeds were found to be very similar with the level of 93.2%. Crossbreeding of these breeds will have no effect on placental characteristics which affect reproduction. Also, placental characteristics may be used for animal selection to increase lambs’ birth weight. For future studies, more breeds and/or other reproductive characteristics can be examined. The decision about the method selection, such as algorithms and distance measures, may be have an important effect on the robustness, so the method selection will need statistical assistance.

## Figures and Tables

**Figure 1 animals-11-00930-f001:**
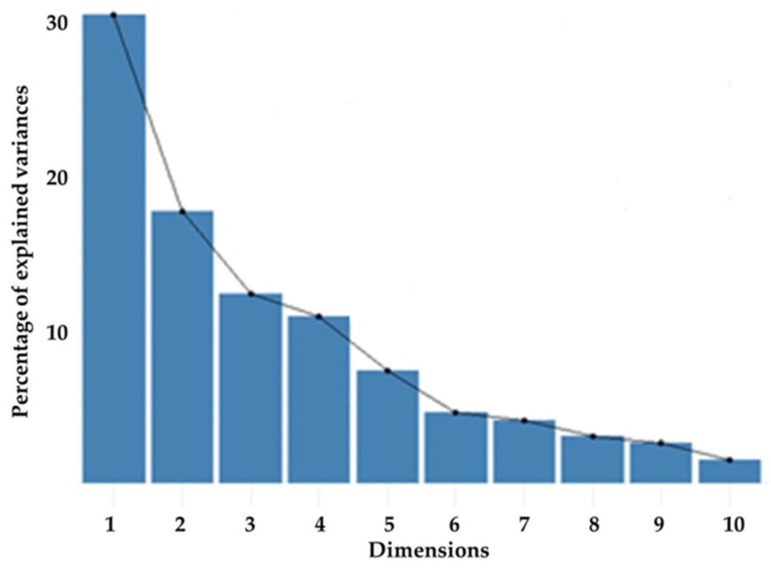
Contribution of components on explained total variance.

**Figure 2 animals-11-00930-f002:**
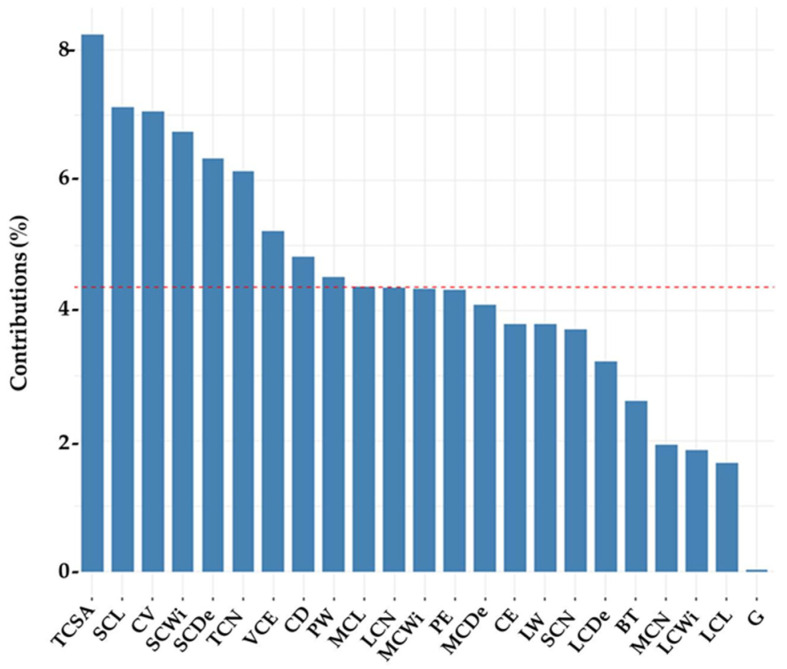
Contribution of variables to components 1 and 2. TCSA = total cotyledon surface area, SCL = small cotyledon length, CV = cotyledon volume, SCWi = small cotyledon width, SCDe = small cotyledon depth, TCN = total cotyledon number, VCE = volumetric cotyledon efficiency, CD= cotyledon density, PW = placental weight, MCL = medium cotyledon length, LCN = large cotyledon number, MCWi = medium cotyledon width, PE = placental efficiency, MCDe = medium cotyledon depth, CE = cotyledon efficiency, LW = litter weight, SCN = small cotyledon number, LCDe = large cotyledon depth, BT = birth type, MCN = medium cotyledon number, LCWi = large cotyledon width, LCL = large cotyledon length, G = gender.

**Figure 3 animals-11-00930-f003:**
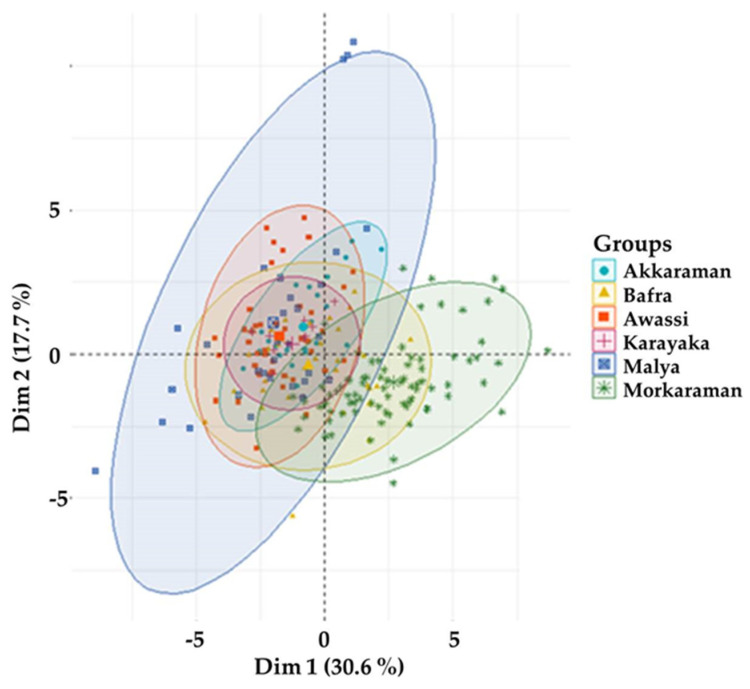
Variance of breeds for contributing on components 1 and 2.

**Figure 4 animals-11-00930-f004:**
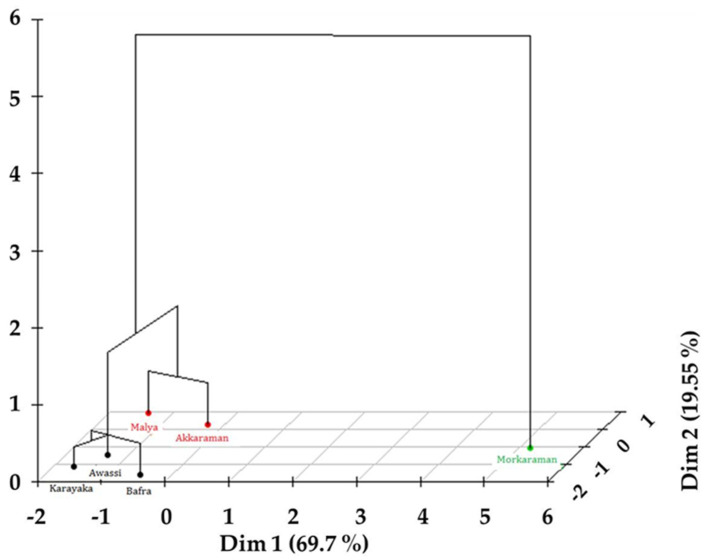
Three-dimensional dendrogram of hierarchical clustering on the factor map.

**Table 1 animals-11-00930-t001:** The reproductive traits of sheep breeds.

Breeds	Litter Size at Birth	Fertility Rate (%)
Akkaraman	1.35 ^a^	89
Morkaraman	1.13 ^b^	92
Karayaka	1.12 ^b^	91
Awassi	1.37 ^a^	89
Malya	1.39 ^a^	88
Bafra	1.15 ^b^	94

a,b: different letters in same row shows the statistical difference (*p* < 0.05).

**Table 2 animals-11-00930-t002:** Descriptive statistics (Mean ± SEM) of placenta and birth-related traits in Akkaraman, Bafra, Awassi, Karayaka, Malya, and Morkaraman sheep breeds.

Variables	Akkaraman	Bafra	Awassi	Karayaka	Malya	Morkaraman
LW	6107.4 ± 436.8 ^b^	4760.3 ± 127.9 ^c^	6116.7 ± 300.7 ^b^	4516.9 ± 141.1 ^c^	6896.1 ± 338.9 ^a^	4582.5 ± 124.9 ^c^
LCN	17.6 ± 2.12 ^a^	10.9 ± 1.29 ^b^	17.05 ± 0.98 ^a^	9.38 ± 0.80 ^b^	16.81 ± 2.79 ^a^	8.70 ± 0.90 ^b^
MCN	26.00 ± 1.06 ^bc^	20.65 ± 1.84 ^c^	29.36 ± 1.91 ^b^	30.46 ± 2.39 ^b^	40.26 ± 4.85 ^a^	44.86 ± 1.79 ^a^
SCN	22.24 ± 2.02 ^ab^	20.48 ± 2.01 ^b^	24.19 ± 1.53 ^ab^	26.38 ± 2.98 ^a^	15.06 ± 1.30 ^c^	3.56 ± 0.59 ^d^
TCN	65.84 ± 2.35 ^ab^	51.93 ± 2.99 ^c^	70.60 ± 3.23 ^a^	66.23 ± 2.72 ^ab^	72.13 ± 7.69 ^a^	57.13 ± 1.72 ^bc^
PW	380.68 ± 24.64 ^b^	258.75 ± 12.86 ^cd^	306.98 ± 16.91 ^c^	230.62 ± 10.44 ^d^	293.71 ± 21.80 ^c^	471.56 ± 14.41 ^a^
LCWi	28.53 ± 0.70 ^a^	27.58 ± 1.23 ^a^	25.09 ± 0.55 ^b^	28.10 ± 0.65 ^a^	23.83 ± 1.06 ^b^	23.03 ± 0.50 ^b^
MCWi	22.84 ± 0.65 ^a^	22.31 ± 0.66 ^a^	18.09 ± 0.44 ^b^	22.54 ± 0.63 ^a^	19.08 ± 0.76 ^b^	22.13 ± 0.47 ^a^
SCWi	15.03 ± 0.52 ^a^	15.49 ± 0.58 ^a^	11.9 ± 0.58 ^b^	13.12 ± 0.36 ^b^	12.14 ± 0.58 ^b^	15.89 ± 0.32 ^a^
LCL	34.10 ± 0.79 ^bc^	38.02 ± 1.62 ^a^	33.64 ± 0.78 ^bc^	36.7 ± 0.79 ^ab^	32.88 ± 1.51^c^	28.26 ± 0.54 ^d^
MCL	26.56 ± 0.57 ^ab^	28.62 ± 0.78 ^a^	24.19 ± 0.49^c^	26.62 ± 0.53 ^ab^	24.88 ± 0.78 ^bc^	27.27 ± 0.59 ^a^
SCL	17.88 ± 0.65 ^b^	20.06 ± 0.59 ^a^	16.26 ± 0.37 ^b^	16.81 ± 0.41 ^b^	17.23 ± 0.60 ^b^	21.13 ± 0.67 ^a^
LCDe	4.37 ± 0.16 ^b^	5.19 ± 0.26 ^ab^	5.66 ± 0.19 ^a^	5.03 ± 0.20 ^ab^	5.01 ± 0.22 ^ab^	5.75 ± 0.27 ^a^
MCDe	3.62 ± 0.17 ^b^	4.94 ± 0.26 ^a^	5.20 ± 0.26 ^a^	4.66 ± 0.26 ^a^	4.56 ± 0.22 ^a^	5.10 ± 0.29 ^a^
SCDe	2.79 ± 0.11 ^b^	3.96 ± 0.24 ^a^	3.16 ± 0.13 ^ab^	3.59 ± 0.16 ^ab^	3.36 ± 0.22 ^ab^	4.01 ± 0.19 ^a^
CV	122.02 ± 10.56 ^b^	117.56 ± 8.12 ^b^	134.76 ± 9.41 ^b^	118.89 ± 5.56 ^b^	147.06 ± 20.45 ^b^	204.64 ± 12.86 ^a^
TCS A	312.37 ± 19.64 ^a^	241.74 ± 13.11 ^b^	261.05 ± 15.09 ^ab^	273.18 ± 14.71 ^ab^	321.33 ± 48.83 ^a^	293.26 ± 11.10 ^ab^
CE	20.01 ± 1.20 ^bc^	22.21 ± 1.54 ^bc^	25.59 ± 1.53 ^b^	17.97 ± 1.27 ^c^	34.43 ± 4.87 ^a^	16.62 ± 0.49 ^c^
CD	0.19 ± 0.01 ^c^	0.22 ± 0.02 ^bc^	0.25 ± 0.01 ^b^	0.30 ± 0.02 ^a^	0.25 ± 0.02 ^b^	0.13 ± 0.01 ^d^
PE	16.37 ± 0.94 ^c^	20.04 ± 1.10 ^b^	21.42 ± 1.09 ^b^	20.21 ± 0.70 ^b^	27.12 ± 2.35 ^a^	10.07 ± 0.29 ^d^
VCE	52.83 ± 2.89 ^b^	47.91 ± 3.69 ^b^	51.61 ± 3.15 ^b^	40.15 ± 2.19 ^bc^	74.42 ± 10.89 ^a^	28.31 ± 1.43 ^c^

a–d: different letters in same row shows the statistical difference (*p* < 0.05). LW = litter weight, LCN = large cotyledon number, MCN = medium cotyledon number, SCN = small cotyledon number, TCN = total cotyledon number, PW = placental weight, LCWi = large cotyledon width, MCWi = medium cotyledon width, SCWi = small cotyledon width, LCL = large cotyledon length, MCL= medium cotyledon length, SCL = small cotyledon length, LCDe= large cotyledon depth, MCDe = medium cotyledon depth, SCDe = small cotyledon depth, CV = cotyledon volume, TCSA = total cotyledon surface area, CE= cotyledon efficiency, CD = cotyledon density, PE = placental efficiency, VCE = volumetric cotyledon efficiency.

## Data Availability

To reach the data please contact with the authors U.S., E.S., S.O., and S.O.Y.

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
