# Peer review of "Placental Characteristics Classification of Various Native Turkish Sheep Breeds"

_animals, 2021, doi:10.3390/ani11040930_

Round 1

Reviewer 1 Report

  1. The language in this manuscript needs further modification and improvement. It is recommended to ask an expert or language editing company to help revise the manuscript. There are grammatical problems, such as: “There are more than 54 million sheep populations with many different local sheep breeds are raised in Turkey.” in Line 45-46.
  2. The author mentioned that placental growth and development its nutrient transfer capacity plays a central role in determination of prenatal growth trajectory of the foetus resulting in alteration of a birth weight. Previous studies reported that lamb survival is highly correlated with birth weight. Then it is recommended to increase the statistics of the survival rate of litter in each breed, conduct a difference test between breeds, and help to select the species recommended for breeding.
  3. Please add species, tissue and period to the title of Table 1. The title of the picture also suggests adding some important information.
  4. Is the variability of these indicators of the uterus large between individuals and between groups? Generally, cluster analysis requires a large variation of indicators.
  5. Figure 3 shows that the genetic relationship between these populations is not far, and the distinction between these populations is not obvious. We should put more energy on choosing breeds with better reproductive performance.

Author Response

Reviewer #1:

We thank reviewer 1 for evaluation of the overall review of the manuscript. The introduction of the manuscript has been improved. Additional descriptive information has been added to the methods section. The improvement has been made in the conclusions.

Comments and Suggestions for Authors:

  1. The language in this manuscript needs further modification and improvement. It is recommended to ask an expert or language................

The language of the manuscript has been checked by Dr. Muhammad Khalid retired from Royal Veterinary College and the University of London, the UK. But if reviewer 1 again thinks it will be better to edit by a professional language editing company, we will do it. Also, introduction section has been revised; please see page 2 lines 47-49, 49-57 and 68-70 in revised MS.

  1. The author mentioned that placental growth and development its nutrient transfer capacity plays a central role in……...

Thank you for your valuable recommendation about survival rate of litter in each breed, but we didn’t emphasize the comparison of the breeds according to their placental characteristics or the effects of placental characteristics on birth weight and survival rate of litter in each breed. Because the aim of the study was classifying sheep breeds using a hierarchical clustering method based on placental characteristics and birth-related traits.

  1. Please add species, tissue and period to the title of Table 1. The title of the picture also suggests adding some important information.

Done as requested, but Table 1 was changed as Table 2 because of the reproductive traits of the breeds were added as Table 1; please see Table 2 in revised MS.

  1. 4. Is the variability of these indicators of the uterus large between individuals and between groups? Generally, cluster analysis…………….

Yes, there are significant variations in terms of placental characteristics among breeds, the differences are shown in table 1 and the classifications were done according to the variation in placental characteristics using a hierarchical clustering method.

  1. Figure 3 shows that the genetic relationship between these populations is not far, and the distinction between these populations is not obvious. We should put more energy.………

We thank much to the reviewer for that good point. That is truth is that the genetic relationship between these populations is not far, but findings were proof of the reliability of clustering results with the use of descriptive statistics.

Reviewer 2 Report

The aim of this study study entitle: „Placental characteristics classification of various sheep breeds” UÄŸur Sen et al., was to classify placental characteristics of Akkaraman, Morkara, Karayaka, Awassi, Malya and Bafra sheep breeds using the hierarchical clustering method.

The authors emphasize that the environmental conditions of pregnant ewes affect the birth weight of their offspring. At the same time, they state that individual herds were kept in different environmental conditions. The description of nutrition is also very general. Please indicate the energy value and protein content of the feed.

Please provide reproductive performance for the breeds you have studied.

In the results and discussions, the authors actually presented only their own results. Out of 19 references, only 6 are the reference to other studies. Please comment whether the differences between breeds were due to the type of birth (singles or twins) or gender (rams have a greater birth weight than ewes).

Line 305  there is:    reproductive performance   should be:  birth weight

Author Response

Reviewer #2:

We thank reviewer 2 for evaluation of the overall review of the manuscript. The introduction of the manuscript has been improved. The conclusions have been improved with some additions.

Comments and Suggestions for Authors:

  1. The authors emphasize that the environmental conditions of pregnant ewes affect the birth weight of their offspring. At the same time, they …….

The paragraph below in the introduction has been revised; please see page 3 lines 122-126 in revised MS.

Old version

“Semi-extensive breeding system was applied in all farms and management and feeding procedures were nearly similar. All breeds were housed and cared for under the similar conditions in the sheepfold (government farms maintain identical conditions and private farms were selected by their similarity of application).”

New version

“All breeds were raised under semi-extensive system and all farms where placentas provided had similar management and feeding conditions. Also, all breeds were housed and cared under similar conditions in the barn (research farms belonging to the state maintain identical conditions, and commercial farms were selected by their similarity of management practices).

Additionally energy value and protein content of the feeds were presented in the MM section. Please see page 3 lines 127-130 in revised MS.

  1. Please provide reproductive performance for the breeds you have studied.

Done as requested, reproductive traits of the breeds, we have studied, presented as a table; please see page 4 lines 88-94 in revised MS. Also, the statistical method for compare reproductive traits of breeds was added to the MM section; please see page 4 lines 71-73 in revised MS.

  1. In the results and discussions, the authors actually presented only their own results. Out of 19 references, only 6 are the reference……..

This clustering study was carried out for the first time in this study based on their placental characteristics and birth-related traits in sheep, and the discussion was formed by evaluating the studies that clustered with different parameters, such as microsatellite markers and body measurements, on the same subject. For this reason, differences among breeds in terms of the type of birth or gender not taken into account directly in the current study.

  1. Line 305 there is: reproductive performance should be:  birth weight

Done as requested, please see page 9 line 407 in revised MS.

Reviewer 3 Report

The manuscript entitled “Placental characteristics classification of various sheep breeds” has been carefully evaluated. In this study, the authors provide data about anatomical properties of the placenta in different Turkish sheep breeds. The MS is written in good language and style, and the result presentation is adequate but the novelty level of study in the relevance for the research field (anatomy, genetics, reproduction) is low since the study is purely descriptive. and does not contain striking new results. There are some limitations of this study due to the fact that ewes should be pregnant in a more synchronized manner, meaning all pregnant ewes which were taking into consideration for this study should be mated in one week. The time range of the whole breeding season is too wide and could have biased fetal and placental development due to environmental/epigenetic factors.  

Author Response

We thank reviewer 3 for evaluation of the overall review of the manuscript. The introduction of the manuscript has been improved. The conclusions have been improved with some additions.

Comments and Suggestions for Authors:

  1. …………. the research field (anatomy, genetics, reproduction) is low since the study is purely descriptive and does not contain striking new results. There are some limitations of this study due to the fact that ewes should be pregnant in a more synchronized manner, meaning all pregnant ewes which were taking into……..

We thank much to the reviewer for that good comment. That is truth is that the time range of the whole breeding season is wide-ranging and could have biased fetal and placental development due to environmental/epigenetic factors, but effect of mentioned factors are minimal since the experimental animals give birth almost at the same breeding season (between September and November) and the environments in which they were bred have similar conditions.

Round 2

Reviewer 1 Report

1.The title should contain country information for these sheep breeds.

2.The author did not provide survival rate information, emphasizing that the main purpose of this study is clustering, but in the introduction, the author said that the purpose of clustering is to better choose breeds that increase productivity and neonatal survival. So, if this research is just clustering, please modify the research purpose in the introduction.

Author Response

Dear Reviewer,

Please find attached the second revised version of our manuscript (animals-1146779) entitled “Placental characteristics classification of various sheep breeds". We would like to thank you for your valuable criticisms and contributions. We have taken account of all the points and have altered our manuscript according to your suggestions. The corrected sentences are red-colored to follow the changes in the manuscript. For your information, the responses to your comments are as follows:

List of Corrections Made in response to the Comments of the Reviewers

Comments and Suggestions for Authors:

  1. The title should contain country information for these sheep breeds

Done as requested, please see page 1 line 2 in R-2 MS.

  1. The author did not provide survival rate information, emphasizing that the main purpose of this study is clustering, but in the introduction………..

Done as requested, please see page 2 lines 54-58 in R-2 MS.

yours sincerely

Dr. UÄŸur ÅžEN

Reviewer 2 Report

After the corrections have been made, I accept the article for publication.

Author Response

Dear Reviewer,

Please find attached the second revised version of our manuscript (animals-1146779) entitled “Placental characteristics classification of various sheep breeds". We would like to thank you for your valuable criticisms and contributions. 

yours sincerely

Dr. UÄŸur ÅžEN

Reviewer 3 Report

No further comments.

Author Response

(The authors gave the same response as above.)
